# Self-Compacting High-Strength Textile-Reinforced Concrete Using Sea Sand and Sea Water

**DOI:** 10.3390/ma16144934

**Published:** 2023-07-10

**Authors:** Vitalii Kryzhanovskyi, Athanasia Avramidou, Jeanette Orlowsky, Panagiotis Spyridis

**Affiliations:** 1Faculty of Architecture and Civil Engineering, TU Dortmund University, 44227 Dortmund, Germany; athanasia.avramidou@tu-dortmund.de (A.A.); jeanette.orlowsky@tu-dortmund.de (J.O.); panagiotis.spyridis@tu-dortmund.de (P.S.); 2Chair of Solid Construction, Faculty of Agricultural and Environmental Sciences, University of Rostock, 18059 Rostock, Germany

**Keywords:** self-compacting concrete, high-strength concrete, textile-reinforced concrete, sea concrete, sea water and sea sand, CO_2_ emission

## Abstract

In this study, a self-compacting high-strength concrete based on ordinary and sulfate-resistant cements was developed for use in textile-reinforced structural elements. The control concrete was made from quartz sand and tap water, and the sea concrete was made from sea water and sea sand for the purpose of applying local building materials to construction sites in the coastal area. The properties of a self-compacting concrete mixture, as well as concrete and textile-reinforced concrete based on it, were determined. It was found that at the age of 28 days, the compressive strength of the sea concrete was 72 MPa, and the flexural strength was 9.2 MPa. The compressive strength of the control concrete was 69.4 MPa at the age of 28 days, and the flexural strength was 11.1 MPa. The drying shrinkage of the sea concrete at 28 days exceeded the drying shrinkage of the control concrete by 18%. The uniaxial tensile test showed the same behavior of the control and marine textile-reinforced concrete; after the formation of five cracks, only the carbon textile reinforcement came into operation. Accordingly, the use of sea water and sea sand in combination with a cement with reduced CO_2_ emissions and textile reinforcement for load-bearing concrete structures is a promising, sustainable approach.

## 1. Introduction

### 1.1. Research Framework

Today, concrete remains the most popular building material in the world [1]. In addition to Portland cement as the main binder, there is about 30–35% sand per 1 m^3^ of coarse-grained concrete, while the amount of sand rises to 65–80% in the production of fine-grained concrete. The minimum amount of water to ensure cement hydration is about 25% of the cement weight [2,3], but this amount of water is not sufficient to ensure the necessary workability of the concrete mixture. Even with the use of modern superplasticizers, the total amount of water in concrete is in the range of 7–12% by volume.

It is worth noting that only about 2.5% of the world’s ocean water is fresh water, and about 40% of the world’s population lives within 100 km of the sea coast [4]; the average distance to the sea coast of the European population is 50 km. Sea coasts also carry a particular socioeconomic significance, as they host approximately 50% of the tourism industry [5], while the maritime transport accounts for more than 80% of international trade volume [6]. Moreover, coastal areas encounter an increased demand for structures for adaptation and defense due to intensifying extreme weather events and sea level rises related to climate change [7,8]. With the active development of the construction sector in coastal areas, the problem of resource-efficient use of sea water and sea sand for the production of concrete arises. Hence, according to the authors, the use of these materials is promising in terms of sustainable construction and energy efficiency in coastal construction. This paper first presents the framework and overall demands for sustainable use of the natural resources in terms of cement production, sand aggregates and water—in particular under the life cycle spectrum of the use of non-metallic/non-corroding reinforcement. It further presents the constituents of a concrete composite element, specially developed by the authors, based on sea water, sea sand and textile reinforcement. Its structural and material properties are characterized based on a series of suitable laboratory tests, and its efficiency is evaluated.

### 1.2. Problem Statement

Today, it is widely recognized that concrete is one of the greatest contributors to global warming, and different strategies to produce environmentally efficient concretes already exist and are implemented in practice. Its environmental sustainability can be assessed on the basis of the European standard EN 15978 [9], and it is determined separately for the operation and construction stages and at different asset life cycle stages (modules). The modules correspond to the manufacturing phase (A1–A3), construction phase (A4–A5), use phase (B1–B7), disposal phase (C1–C4) and the potential for reuse (D). Although various measures can be implemented to counter the environmental/climate impacts of construction, greenhouse gas emissions are represented herein as based on their equivalence with CO_2_ in terms of climate impact. This is a measurement scale developed and implemented by the United Nations’ Intergovernmental Panel on Climate Change [10] to represent the influence of various gases on global warming. CO_2_ equivalents are also referred to as global warming potential (GWP) and abbreviated to CO_2_-eq in mass units (kg). For a product (including construction products, such as structural concrete constituents), these values can be found in its respective environmental product declaration (EPD), which is defined by ISO 14025 [11]. The common product category rules (PCR), i.e., the guidance for EPD development, are outlined for the construction sector in Europe by EN 15804 [12].

Lately, the material technology has offered various possibilities for the replacement of cement with other binders, such as calcium sulphate aluminates (CSA) and geopolymers, which can lead to a relative reduction of approximately 30% or 80%, respectively [13]. Replacing the Portland cement clinker content with other so-called secondary cementitious materials (SCM) would be a recommended method to reduce concrete’s GWP (Table 1). These materials include fly ash, granulated blast furnace slag (GBFS), limestone, pozzolans and silica fume. Pure Portland cement CEM I with a clinker content between 95 and 100% was considered the most commonly used cement type, but its use has decreased in recent years due to its very environment- and energy-intensive production [14]. The four other main types of cement include Portland composite cement CEM II, blast furnace cement CEM III, pozzolanic cement CEM IV and composite cement CEM V. As seen in Table 1, CEM III stands out as the one with the lowest GWP due to its lower clinker content and primary energy consumption. Another advantage of CEM III includes its resistance to chemical attack due to its finer pore structure, which also renders it suitable for use in marine environments [15]. Moreover, CEM III has a lighter color tone, which leads to esthetically superior fair-faced concrete solutions. Finally, GBFS is expected to remain available in the future unlike other binder replacements, such as fly ash, which are derived from declining industry practices. 

Sand has for a long time been considered an easily accessible and seemingly abundant raw material, which can be extracted from mines and deposits, including deserts. Vast quantities, in the range of 50 billion tons annually, are consumed globally in the production of various industrial products, such as glass, filters, microchips/electronics, shale oil from fracking, but also in construction as a filler or a concrete aggregate [16]. In recent years, however, the depletion of sand from over-extraction (often also illegal or unregulated) has led to grave environmental impacts, such as the destruction of flora and fauna habitats, loss of natural floods and erosion protection, or disruptions to tourism [17]. Moreover, only 5% of the available sand is suitable for use as a concrete aggregate, which requires a certain roughness and toughness to allow cohesion with the binder matrix. Sea sand fulfills these requirements, but it cannot be used immediately, as it must be desalinated in order to avoid corrosion of standard steel reinforcement. This process requires approximately 2 tons of fresh water and 0.52 kWh of energy per ton of sea sand [18]. Lastly, sand dredging from the sea bottom poses a significant efficiency problem in terms of emissions, with 0.105 kg CO_2_,e/ton compared to the extraction from pits and quarries with 0.276 and 0.737 kg CO_2_,e/ton, respectively [19]. 

Regarding water, as another significant constituent of concrete, out of the 1.4 billion km^3^ of water on earth, only 2.5% is drinkable fresh water, 0.3% of which is directly accessible via rivers, streams or lakes on the continental surface [20]. At the same time, drinking water scarcity is an intensified issue globally, with the shortage being either due to lack of a water supply infrastructure or lack of running water and rainfalls leading to drought. Globally, agriculture accounts for 70% of water consumption, followed by industry at 20% and municipal consumption at 10%. Currently, approximately 3 to 5 billion tons of water is used for concrete production, while a projection for the year 2050 by [21] indicates that water demand for concrete production will be approximately 75% for regions experiencing drought stress. Considering this situation, the use of sea water for concrete production can be vital to the conservation of this vital resource in future construction, which in fact has already been used in the Roman era for this reason and has historically proven to create a very durable building material [22]. As with sea sand, however, sea water inherently incorporates chloride ions in the concrete leading to corrosion issues for steel reinforcement.

### 1.3. The Ocean as a Source of Basic Concrete Constituents

In experiments [23,24,25,26], it has been observed that sea concretes obtained equal strength compared to control concretes with normal river sand and tap water. The compressive strength of the sea concretes was in the range of 30–80 MPa, depending on the composition. In experiment [27], concretes with compressive strengths from 28 to 38 MPa were obtained after 180 days of hardening with different types of cements, but studies also indicated overall accelerated setting times due to the salts. In experiment [28] confirmed the possibility of obtaining ultra-high-strength sea concretes with a compressive strength over 100 MPa and flexural strength over 15 MPa. The microstructure of concrete with sea sand and sea water shows a denser structure, which has a positive effect on concrete durability [29]. Study [30] found that sea concrete had high water tightness and frost resistance due to its denser structure compared to conventional concrete. Shrinkage deformations of sea concrete tend to increase, as described by the authors [31,32,33,34].

It should be noted that for construction in the coastal zone, it is important to ensure sufficient sulphate resistance of concrete structures, according to [35]. A separate problem is the possibility of using steel reinforcement in concretes with sea sand and water due to the increased risk of corrosion. In this regard, textile-reinforced concrete can be an effective solution. In addition to the high strength of textile reinforcement (basalt, glass, carbon), it has increased resistance to aggressive environments [36,37]. For additional sulphate resistance of the concrete matrix, sulphate-resistant Portland cement [38] can be used. The use of textile-reinforced concrete also makes it possible to significantly reduce the thickness of structures, which has a positive effect on saving resources during their manufacture.

Textile-reinforced concrete (TRC) contains a rectangular arrangement of fiber bundles made of non-corroding materials, such as carbon, glass or basalt. Textile reinforcement is produced from continuous fibers (filaments) in the form of rovings, which can contain more than 40,000 individual filaments. Textile reinforcement has close-meshed openings, which increase the ductility of concrete. Stresses and cracks are distributed over a larger area, thereby reducing cracking in the structure elements. The smaller diameter of the cross-section of textile reinforcement (1–2 mm) allows to reduce the thickness of structures and consequently reduce their total weight [39,40]. A feature of the production of TRC is the need to use fine-grained concrete, since the step of reinforcing mesh often does not exceed 1.5–2.5 cm. In addition, the concrete must have the necessary workability for uniform distribution in the formwork with dense textile reinforcement, as well as high strength to ensure the effective function of textile strings in the concrete [41,42]. As is known from global building practices, self-compacting concrete is used for the concreting of densely reinforced structures [43,44,45,46,47]. Consequently, the development of self-compacting, high-strength concrete using sea water, sea sand and sulphate-resistant cement is an urgent task. The main objectives of this study are to determine the effect of sea water, sea sand and different types of cement on the flowability and air content of self-compacting concrete mixtures and to determine the strength, water absorption and drying shrinkage of concrete and textile-reinforced concrete based on them. Special attention is devoted to the possibility of reducing CO_2_ emissions through the use of sea textile-reinforced concrete.

## 2. Materials and Methods

Two types of cement were used to prepare self-compacting concretes: Dyckerhoff CEM I 42.5 N (bulk density = 1310 kg/m^3^) for the control concrete mixture and Dyckerhoff CEM III/B 42.5 N-LH/SR (bulk density = 1180 kg/m^3^) for the sea concrete mixture. The use of CEM III/B 42.5 cement results in a 47% reduction in CO_2_ emissions (CEM I 42.5 N/CEM III/B 42.5 N-LH/SR = 661/311 kg CO_2_-Eq/t). Normal and sea sand were used as fine aggregates; the particle size distribution was determined according to Ref. [48] and shown in Figure 1. The sea sand comes from Katerini, Greece. The density of the sea water was recorded with a salinity meter to determine the salinity. The density is ρ = 1025 kg/m^3^; using https://reefapp.net/de/salinity-calculator (accessed on 23 April 2023), this corresponds to a salinity of 3.5%.

The bulk density and water absorption of the sands were determined in accordance with [49]; the results are shown in Table 2. Millisil W12 quartz flour with maximum grain size 50 µm and bulk density 900 kg/m^3^ was used as a microfiller. Polycarboxylate-type superplasticizer MasterGlenium ACE 460 was used as a water-reducing admixture. Tap water and sea water for aquariums were used to manufacture the control and sea self-compacting concrete mixtures.

Based on the compositions of self-compacting concretes given in Table 3, two batches of prism samples with dimension 4 × 4 × 16 cm were produced (18 samples in each batch), according to the standard [50]. Cement, sand and the microfiller were mixed for 3 min; then, water with the superplasticizer was added, and the mixing continued for a further 5 min. The fresh concrete was cured in accordance with [50]. The samples were demolded after 24 ± 2 h; then, the samples were placed in a climatic chamber (t = 20 ± 2 °C, relative humidity 65 ± 5%).

The flowability of the investigated self-compacting concrete mixtures was determined by the slump flow test [51]. Figure 2 shows the resulting self-compacting concrete mixtures. The density of fresh concrete mixtures was also determined according to [52], and the amount of entrained air was measured (Figure 3) according to [53]. The results are shown in Table 4.

To study the mechanical properties of textile-reinforced concrete (TRC), 2 plates with a size of 1300 mm × 420 mm × 10 mm were produced based on the compositions of self-compacting concrete from Table 3. Carbon mesh produced by V. Fraas with 25.4 cm × 25.4 cm spacing was used as textile reinforcement. The cross-sectional area of one strand was 1.81 mm^2^, the central tensile strength was 3155 MPa, and the modulus of elasticity was 220 GPa.

First, a 6 mm layer of concrete was placed; then, the carbon mesh was laid and slightly troweled; then, a second layer of 4 mm concrete mixture was placed and then troweled (Figure 4). The manufactured plates were covered with polyethylene film to prevent moisture loss. Demolding was carried out 24 ± 2 h after the samples’ production. Afterward, the plates were stored in a climatic chamber (t = 20 ± 2 °C, relative humidity 65 ± 5%). Seven days before the uniaxial tensile test, specimens with dimensions of 1050 mm × 70 × 10 mm were cut from one plate, so that one specimen contained three carbon rovings (Figure 5). After cutting, the specimens were placed back in the climatic chamber until testing.

The following physical and mechanical properties were determined for the investigated self-compacting fine-grained concretes and textile-reinforced concretes based on them:-compressive strength at age of 3, 7 and 28 days according to the methodology in [54], load application rate 2.45 kN/s;-flexural strength at age of 3, 7 and 28 days according to the methodology in [54] (three-point load schema), load application rate 0.05 kN/s;-modulus of elasticity at the age of 28 days [55];-drying shrinkage determined with linear method [56];-water absorption: the samples were initially dried for 24 h at 105 °C, then weighed. Afterward, the samples were placed in a water bath for 48 h in such a way that the water level was 5 cm above the top of the samples. The samples were then taken out from the water bath, lightly wiped with a dry cloth and weighed. The water absorption was determined in %, as the mass difference between a dry and a water-saturated sample;-uniaxial tensile strength test [57], load rate 1 mm/min.

## 3. Test Results and Discussion

### 3.1. Compressive Strength and Water Absorption

The density of control and sea self-compacting concretes was in the range of 2220–2240 kg/m^3^. Figure 6 shows a diagram of the compressive strength of self-compacting concretes at different ages.

The water absorption of control concrete was 5.5%, and the water absorption of sea concrete was 5.1% (three samples for each batch). This may be due to the high air content of the fresh concrete mixture and the consequent formation of more open pores on the surface of the hardened concrete. Additionally, refs. [58,59,60] confirmed the positive effect of sea mixing water on the pore structure of concrete, namely the reduction in capillary pores. According to [61], the use of sea sand also improves the pore structure of concrete, which has a positive effect on its durability.

Six samples for each batch were used for compressive strength tests. The compressive strength of the control self-compacting concrete at the age of 3 days was expectedly higher than the compressive strength of sea self-compacting concrete; the difference was 29.2%. This is due to the fact that the sea concrete was made on a CEM III/B 42.5 N-LH/SR base with a low clinker content, and consequently, a lower content of tricalcium aluminate C_3_A, which contributes significantly to early strength development. At the age of 7 days, the difference in compressive strength decreased to 2%—65 MPa for control concrete and 63.7 MPa for sea concrete. At the design age, the strength of sea concrete exceeded the strength of control concrete by 3.7% and amounted to 72 MPa. 

The authors in [62,63,64] confirm the positive effect of using sea sand and sea water in concrete production. The strength at the age of 28 days in all cases corresponded to the design strength of the concretes. Separately, it should be noted that early concrete strength at the age of 3 days is not a decisive indicator in new construction. Thus, in our experiment, self-compacting high-strength concretes were obtained (SCHSC). 

### 3.2. Flexural Strength

The data obtained on the flexural strength of the studied self-compacting concretes are shown in the diagram in Figure 7 (three samples for each batch). At the age of 3 days, the flexural strength of the control concrete was 7.1 MPa, and that of sea concrete was 6.1 MPa, which was 16% less. This result is explained by the reduced strength of the cement matrix based on slag cement at an early age, as described above. In turn, on the seventh day of hardening, the flexural strength of sea concrete exceeded the flexural strength of the control concrete by 1.3% and reached a value of 7.7 MPa. On day 28, the flexural strength of the control concrete composition was 11.1 MPa, and that of the sea concrete was 9.2 MPa. This is due to the different quality of the interfacial transition zone (ITZ) of the fine aggregate and cement. The ITZ of sea sand with cementitious matrix is weak due to the smooth grain surface [23,65], and the additional sites of tensile stress concentration are porous coral particles and shells [66,67].

### 3.3. Elastic Modulus

The modulus of elasticity of concrete is an important parameter, which is used in the calculation of building structures and characterizes the ability of concrete to retain its elastic properties under external load. Figure 8 shows concrete specimens before testing (three samples for each batch).

For sea concrete, the modulus of elasticity E_c_ at the age of 28 days was 39.1 GPa, and for the control concrete, it was 38.9 GPa. This is due to the lower compressive strength of the control concrete composition. Other studies [26,68] also found that sea sand and sea water do not affect the modulus of elasticity of concrete, and only concrete strength affects this indicator; higher compressive strength leads to an increase in the modulus of elasticity.

### 3.4. Drying Shrinkage

Shrinkage is one of the most important parameters for the durability of concrete. In this study, the drying shrinkage was determined using the linear method (Figure 9; three samples for each batch). Figure 10 shows the dynamics of the development of shrinkage deformations over 56 days of hardening. It should be noted that the shrinkage deformation of sea concrete at the early age of 3 days was 0.6 mm/m, which exceeded that of control concrete by 122%. This was expressed due to the lower strength of the sea concrete composite at an early age, since the action of capillary forces exceeded the tensile strength and caused higher deformations in the weak concrete matrix of the sea concrete compared to the control concrete. On the seventh day of hardening, the shrinkage deformation of sea concrete was 0.78 mm/m, while that of control concrete was 0.55 mm/m, which was 42% lower. At the design age, the shrinkage of sea concrete exceeded the shrinkage of control concrete by 18% and amounted to 1.02 mm/m. After 56 days of hardening, the shrinkage of the control concrete was 0.97 mm/m, and the shrinkage of the marine concrete was 1.07 mm/m. The data obtained are consistent with the studies [31,32], which emphasize the increase in shrinkage of concrete with mixing sea water due to the increase in gel pores. In refs. [33,34], the possibility of increasing the concrete shrinkage strain by using sea sand due to the creation of a heterogeneous pore structure is noted.

### 3.5. Textile-Reinforced Concrete—Uniaxial Tensile Test

On the basis of SCHSC control composition and SCHSC sea composition, samples reinforced with carbon mesh were manufactured (three samples for each batch). The tensile tests were carried out according to the load application scheme shown in Figure 11. Figure 12 shows the stress–strain diagram of the investigated TRC.

The diagram shows that the tensile strength of sea concrete was 3.6 MPa, and the tensile strength of control concrete was 3.3 MPa. This area on the diagram has a linear character. With the development of the first crack, textile reinforcement is included in the work, and the stage of multiple cracking begins—a sawtooth section in the diagram. It was experimentally established that the number of cracks on both types of concrete was five pieces. The average distance between cracks on the samples of sea TRC was 9.6 cm, and on the samples of control TRC, the distance was 10 cm. With a further increase in the load, the bond between the reinforcing mesh and concrete is broken, and only textile reinforcement continues to work.

The absorption energy of sea and control TRC was also calculated for specimen displacement at 10 mm. For this purpose, the average area (three samples) under the stress–strain curve for each TRC type was determined. For the sea TRC composition, the fracture energy was 874.56 kJ, and for the control TRC composition, it was 835.12 kJ. Thus, 4.7% more energy was required to achieve a displacement length of 10 mm for the sea TRC compared to the control TRC.

The obtained data on the performance of textile-reinforced concrete are consistent with the research [41,42,69]. Since the strength characteristics of sea and control concretes are of equal limits, the effect of textile reinforcement is equal. In general, the use of sea SCHSC as the main matrix for the production of textile-reinforced structures is promising in terms of reducing the material consumption compared to classically reinforced concrete.

### 3.6. CO_2_ Emissions of SCHSC

Since the purpose of this study was to determine not only the physical and mechanical properties of concretes but also their impact on the environment, this section determined the amounts of CO_2_ emissions per 1 m^3^ of concrete in the extraction and production stages (A1–A3), according to [9], which are shown in Figure 13. All data for the GWP calculations were taken from [19,70,71,72,73,74,75].

The data show that the use of Type III low clinker cement, sea sand and sea water for SCHSC production can reduce CO_2_ emissions by 91.5% compared to the control SCHSC. In turn, the use of slag cement has the most impact on decreasing CO_2_ emissions. Additionally, the use of sea sand up to 50 km from the coast would generate much less emissions than the transportation of sand from a quarry.

A direct comparison between steel reinforcement and textile reinforcement with carbon rovings via their global warming potential does not seem appropriate due to an energy-intensive process. However, it should be considered that carbon reinforcement is much lighter in weight, in addition to the fact that the bearing capacity can be up to six times higher [63]. There is no need for a sufficient protective layer of concrete, which should be at least 20–50 mm for corrosion protection. For this research, the concrete cover is barely 4 mm thick, which corresponds to a reduction of 20%. Thus, it is possible to reduce the complete cross-sectional thickness of structural elements, which can save up to 80% of their material consumption [76]. Over the entire life cycle and durability, low specific CO_2_ emissions are recorded at the component level compared to steel in the following research works [77,78,79].

## 4. Conclusions and Outlook

This paper provides an overview of the use of sea water and sea sand in combination with textile reinforcement with the following conclusions:-the flowability of the sea concrete was more viscous than the control concrete because of the amount of sea shells, which are contained in the sea sand, and because of the salt in the sea water, which accelerated the hydration. Therefore, a higher content of superplasticizer for the concrete mixture with sea water and sea sand can be expected. Additionally, the air content test confirmed that, although self-compacting concrete was produced, which usually has a higher air content, in this case, it was below the limit of 5–6% and had no negative influence on the pore structure;-the use of sea sand and sea water for the production of SCHSC based on slag cement makes it possible to produce concretes with comparable mechanical properties while significantly improving the life cycle assessment. At the age of 3 days, the compressive strength of the sea concrete composition was 37.3 MPa, which was around 10.9 MPa lower than the compressive strength of the control concrete composition at 48.2 MPa, due to the slower hydration rate of the slag cement. However, this cannot be considered a negative effect because, in new construction, early strength is not paramount. It was experimentally established that at the age of 28 days, sea concrete had a compressive strength of 72 MPa, which corresponds to the concrete grade of C45/55. The strength of the control concrete was 69.4 MPa, which corresponds to the concrete grade of C40/45. The flexural strength at the age of 3, 7, 28 days of sea concrete was slightly lower (6.1 MPa, 7.7 MPa, 9.2 MPa) than the flexural strength (7.1 MPa, 7.6 MPa, 11.1 MPa) of control concrete due to the weak surface contact between the shells and the cement matrix;-after 28 days, the modulus of elasticity for the sea concrete showed a higher value of 39.1 GPa in comparison to the control concrete value of 38.9 GPa, which showed that sea water had no effect on it;-the studies on durability in the form of drying shrinkage show that at the early ages (3 days), the sea concrete shrinkage deformations exceed 122% with 0.6 mm/m in comparison to the control concrete. This is explained by the fact that, with the use of sea water, the pore structure is refined, and the capillary pores (>100 nm) are reduced [18]. Due to their size, the capillary pores have a faster and better water transport. At the same time, the number of fine gel pores increases, but they can hold the water for a very long time and cause larger shrinkage deformations [80]. After 56 days, the shrinkage deformations between the sea concrete (1.07 mm/m) and the control concrete (0.97 mm/m) had a fading character;-the combination effect of sea concrete as an SCHSC with textile reinforcement on the tensile strength (3.6 MPa) shows equal results to the control concrete (3.3 MPa) with a slightly increasing trend. The obtained data on the performance of textile-reinforced concrete based on sea and control concrete matrices show the same result.-the raw materials used for the production of concrete in our experiment can be recommended for the construction of facilities near the sea coast, which makes it possible to obtain SCHSC. It is important to note that by using slag cement in the production of textile-reinforced structures, CO_2_ emissions can be significantly reduced. The ability to use sea sand as an aggregate also eliminates the need to deliver this material to the construction site, which further reduces CO_2_ emissions. Figure 13 shows that the decrease in CO_2_ emissions for 1 m^3^ of concrete would be more than 90% in total. In addition, the use of textile-reinforced concrete allows us to reduce the cross-sectional thickness of structural elements, which makes it possible to reduce the material consumption and bypass corrosion of steel reinforcement due to sea sand/water.

Further research should be directed to the study of the microstructure of sea concretes. Of particular interest is the study of the effect of different salinities of sea sand and sea water on the properties of sea concrete. The possibility of reducing shrinkage deformations of sea concrete by using dispersion reinforcement must also be considered. Special attention should be paid to durability indicators, such as frost resistance, sulphate resistance and water permeability. This will expand the existing data and, in the future, will allow the use of concrete based on slag cement, sea water and sea sand not only for residential construction in the coastal zone but also for the construction of coastal protection and offshore structures.

## Figures and Tables

**Figure 1 materials-16-04934-f001:**
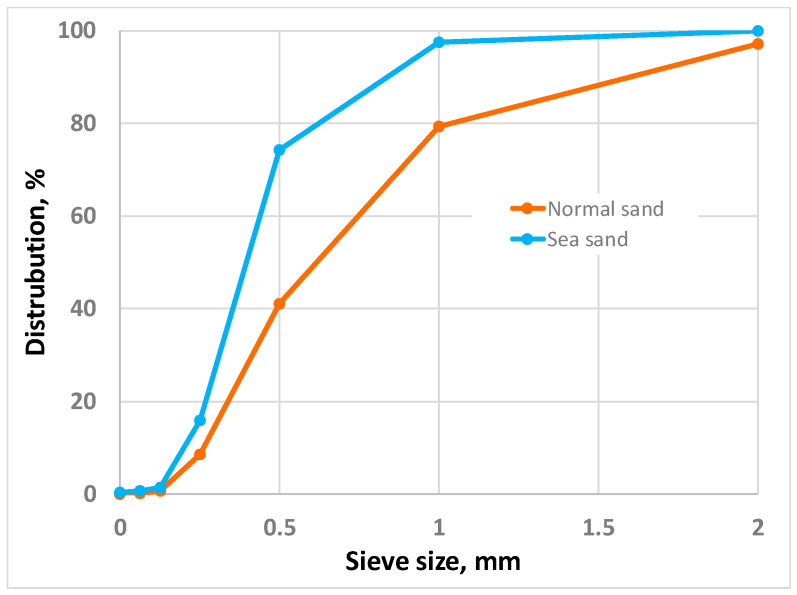
Sand particle size distribution.

**Figure 2 materials-16-04934-f002:**
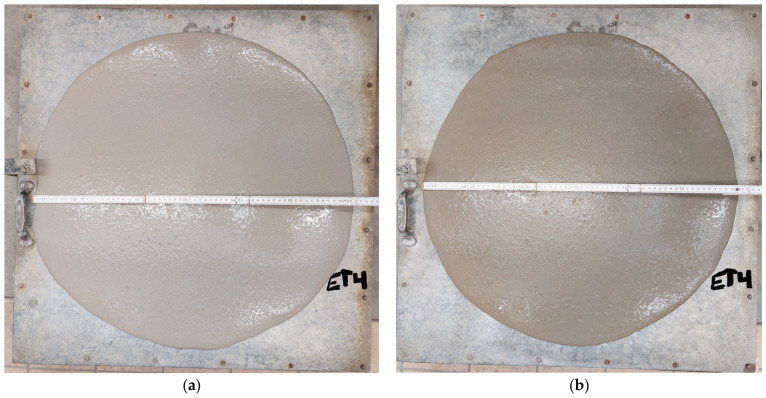
Flowability of self-compacting concrete mixtures: Sea concrete mixture (**a**), control concrete mixture (**b**).

**Figure 3 materials-16-04934-f003:**
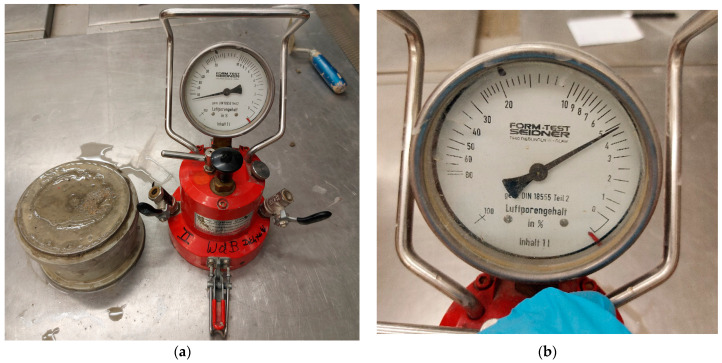
Air content test: Concrete mixture after testing (**a**), instrument readings (**b**).

**Figure 4 materials-16-04934-f004:**
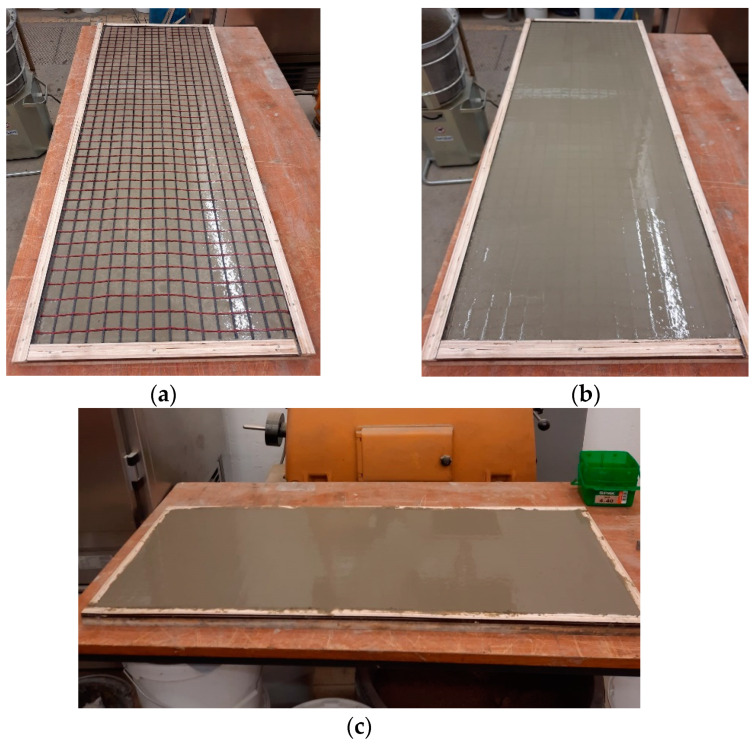
Production process of textile-reinforced plate: First concrete layer of 6 mm and carbon mesh (**a**), troweled carbon reinforcement (**b**), finished plate (**c**).

**Figure 5 materials-16-04934-f005:**
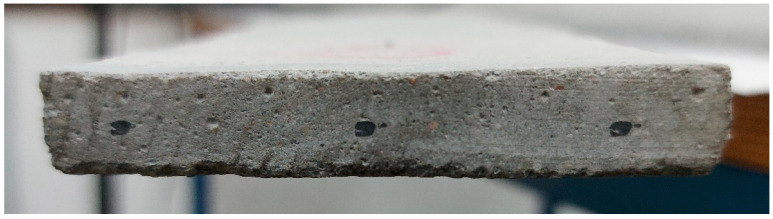
Cross-section of the textile-reinforced specimen.

**Figure 6 materials-16-04934-f006:**
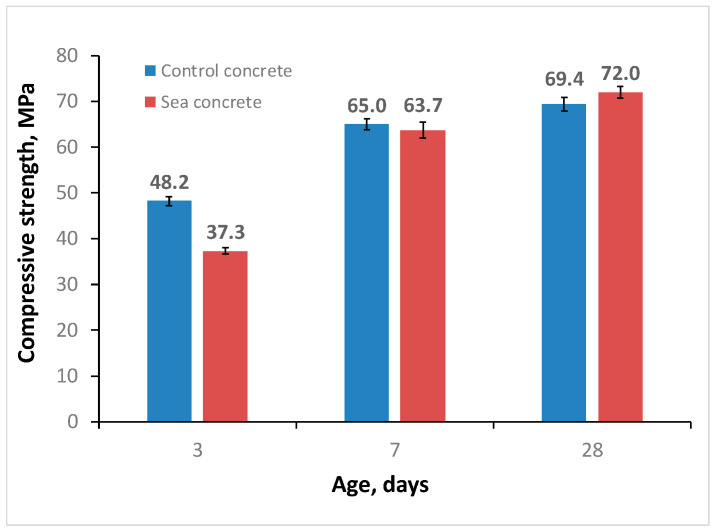
Compressive strength of self-compacting concretes.

**Figure 7 materials-16-04934-f007:**
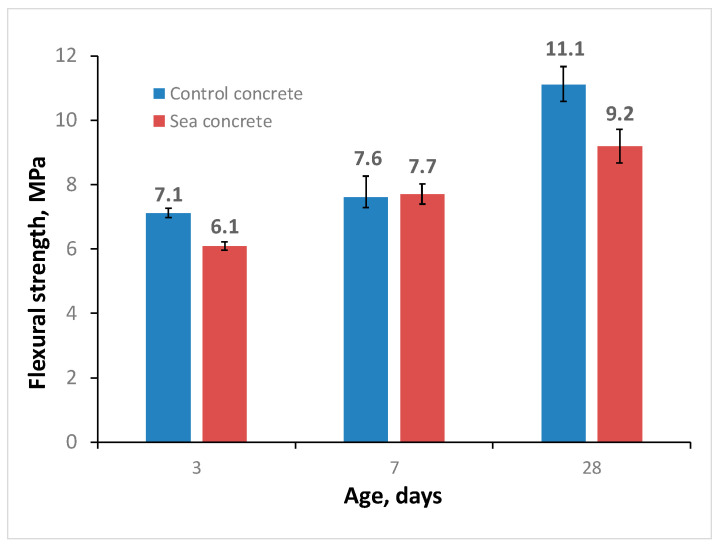
Flexural strength of self-compacting concretes.

**Figure 8 materials-16-04934-f008:**
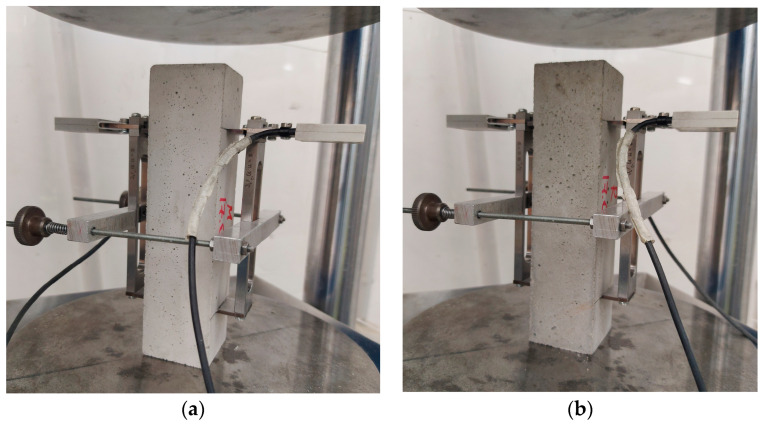
E-modulus test: Sea concrete (**a**), control concrete (**b**).

**Figure 9 materials-16-04934-f009:**
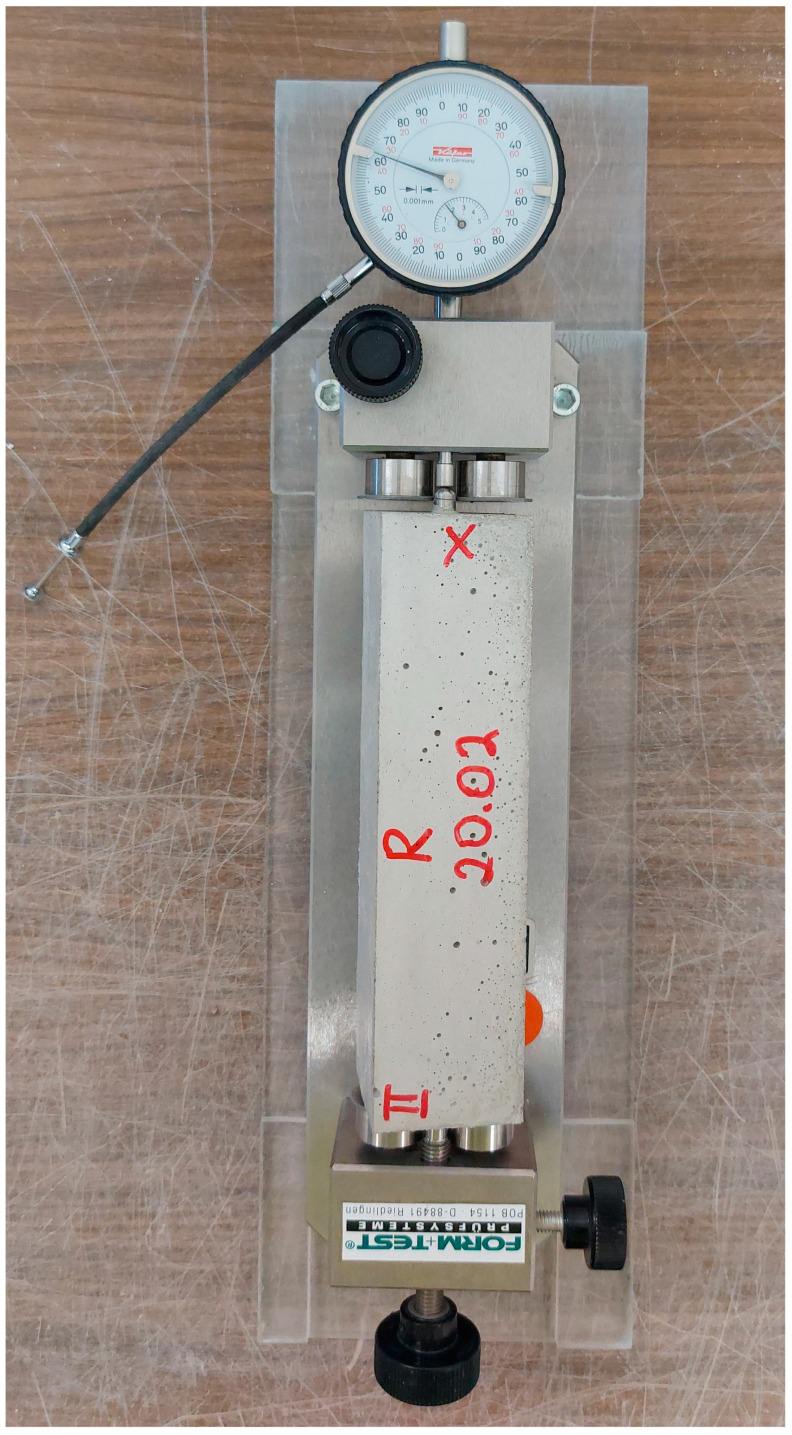
Concrete shrinkage test.

**Figure 10 materials-16-04934-f010:**
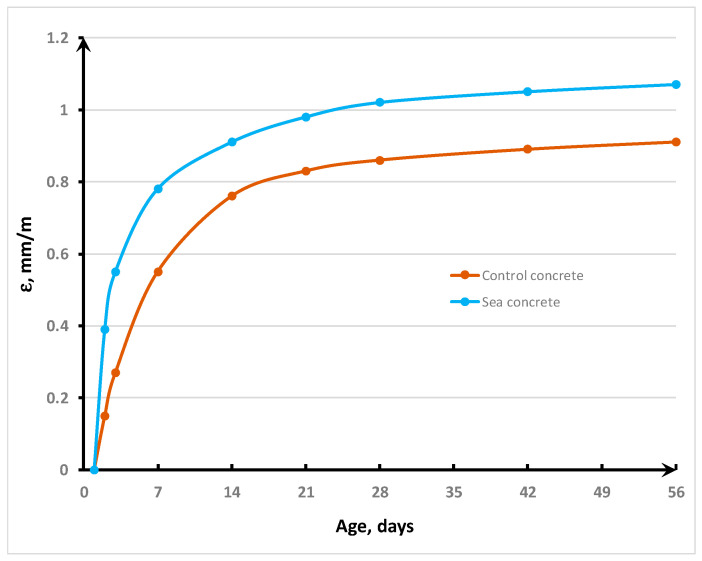
Drying shrinkage of the studied SCHSC.

**Figure 11 materials-16-04934-f011:**
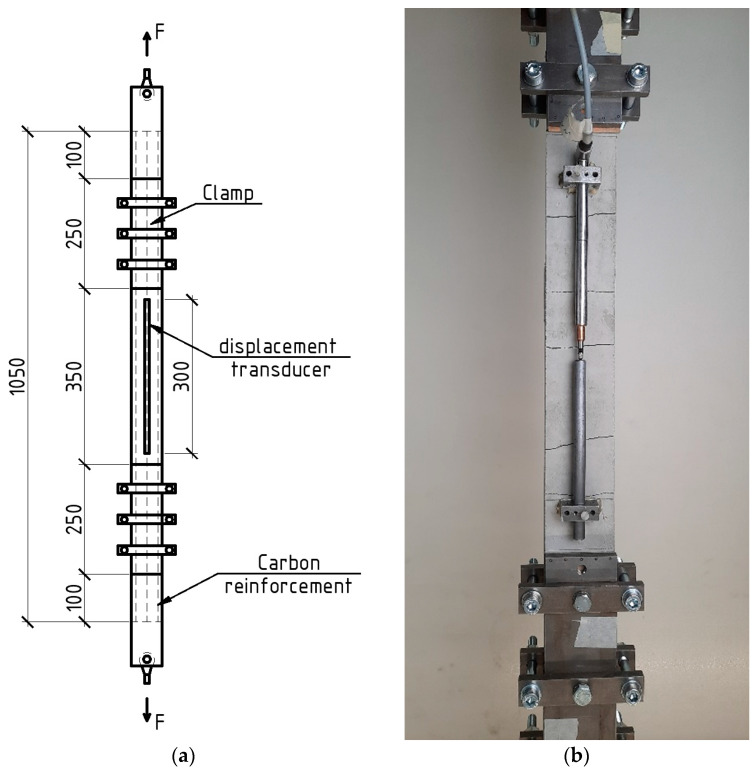
Uniaxial tensile test: Load scheme (**a**), textile-reinforced specimen after test (**b**).

**Figure 12 materials-16-04934-f012:**
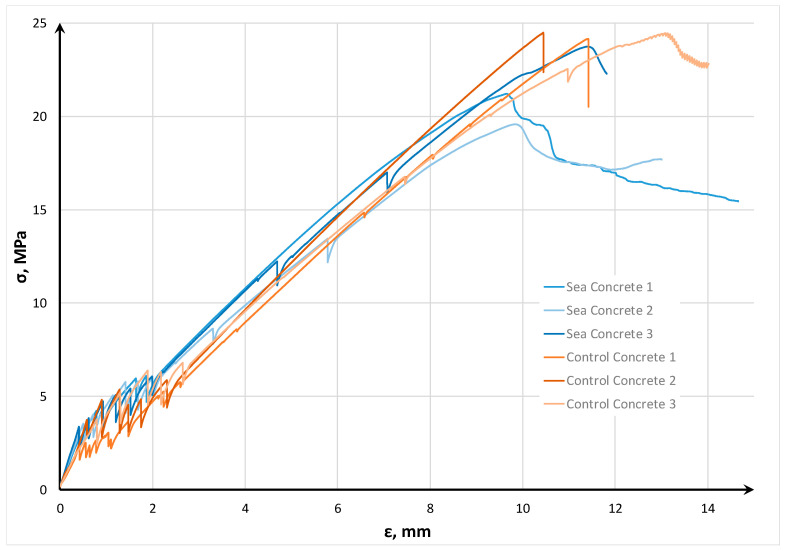
Stress–strain diagram of the investigated textile-reinforced concretes (TRC).

**Figure 13 materials-16-04934-f013:**
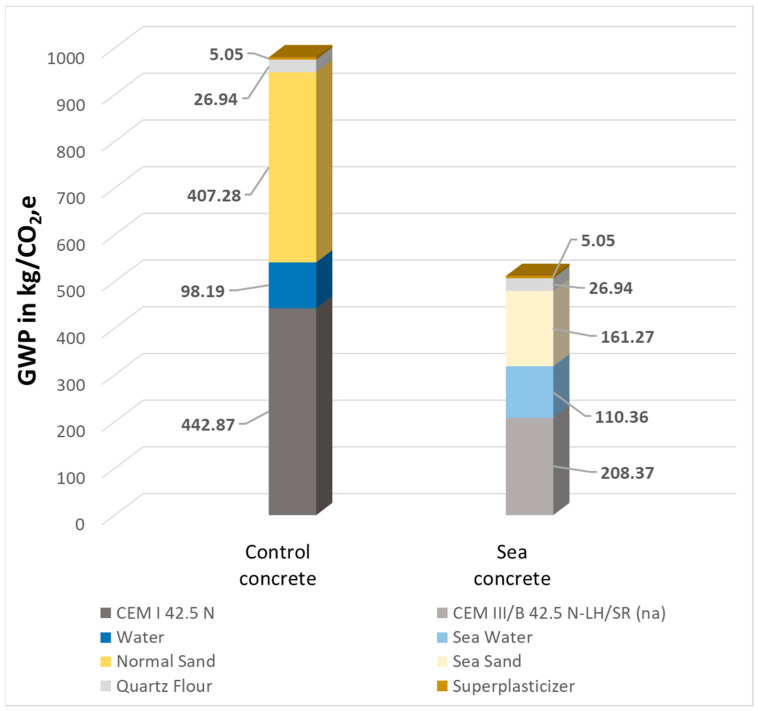
CO_2_,e emissions of 1 m^3^ concrete for the materials’ production phase A1–A3 [9].

**Table 1 materials-16-04934-t001:** Global warming potential (GWP) in CO_2_ equivalents for various cement types and binder replacements.

	GWP in kg CO_2_,e/kg
Carbon Dioxide (CO_2_)	1
CEM I 32.5 (Portland cement)	0.664
CEM I 42.5 (Portland cement)	0.797
CEM I 52.5 (Portland cement)	0.816
CEM II 32.5 (Portland composite cement)	0.785
CEM II 42.5 (Portland composite cement)	0.794
CEM II 52.5 (Portland composite cement)	0.807
CEM II/A	0.874
CEM II/B	0.718
CEM III 42.5 (GBFS)	0.386
CEM III 52.5 (GBFS)	0.399
CEM III/B 42.5-LH/SR	0.281
CEM IV 32.5 (Pozzolan cement)	0.674
CEM IV 42.5 (Pozzolan cement)	0.685
Reference average cement (indicative)	0.553
Fly ash (with allocation)	0.350
Granulated blast furnace slag (with allocation)	0.114
Volcanic ash	0.026
Calcinated clay	0.125
Silica dust	0.028

**Table 2 materials-16-04934-t002:** Sand properties.

Sand Type	Bulk Density, kg/m^3^	Water Absorption, %
Normal sand	1590	2.58
Sea sand	1510	3.51

**Table 3 materials-16-04934-t003:** Compositions of the studied self-compacting concretes.

Mixture	Cement, kg/m^3^	Normal Sand, kg/m^3^	Sea Sand, kg/m^3^	Quartz Flour, kg/m^3^	Superplasticizer, kg/m^3^	Water, kg/m^3^	Sea Water, kg/m^3^	W/C
Control concrete	670	1284	-	224	3.3	268	-	0.4
Sea concrete		-	1284		3.58	-	268	

**Table 4 materials-16-04934-t004:** Fresh concrete properties.

Mixture	Slump Slow, mm	Density, kg/m^3^	Air Content, %
Control concrete	610	2315	4.8
Sea concrete	620	2305	4.6

## Data Availability

The data supporting this study are available from the author on request.

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
