# Peer review of "Self-Compacting High-Strength Textile-Reinforced Concrete Using Sea Sand and Sea Water"

_materials, 2023, doi:10.3390/ma16144934_

Round 1

Reviewer 1 Report

 The paper presents a self-compacting high-strength concrete based on ordinary and sulfate-resistant cements for use in textile-reinforced structural elements. The control concrete was made from quartz sand and tap water, and the sea concrete was made from sea water and sea sand for the purpose of applying local building materials to construction sites in the coastal area. The properties of a self-compacting concrete mixture, as well as concrete and textile-reinforced concrete based on it, were determined. The results showed that the use of seawater and sea sand in combination with a cement with reduced CO2 emissions and textile reinforcement for load-bearing concrete structures is a promising, sustainable approach. Overall, the topic and the research results of the paper is very interesting. Besides, the article is well organized and its presentation is good. However, some minor issues still need to be improved: (1) The pictures in the paper should be adjusted to adapt the article and facilitate the readers. (2) The presentation of the conclusions should be more focused.

The presentation of the conclusions should be more focused and some spelling and grammar errors in the paper need to be corrected.

Reviewer 2 Report

The authors performed an experimental; program to determine the effect of sea sand and seawater on the mechanical properties of textile-contained SCC mixtures. Although the manuscript is interesting, a major revision is necessary, as follows:

-          Page 3, Lines 104-107: Please use a reference for this sentence.

-          Please list this study's main objectives at the end of the introduction.

-          Page 4, Lines 137-140: The authors should consider that replacing reinforcement entirely with textile is not possible in reinforced concrete members containing sea sand and water to solve the corrosion phenomenon.

-          Section 2: Please mention the densities of types of cement.

-          Table 3: was constant SP used for both mixtures? It can not be possible to have a similar slump in these mixtures without changing the SP. Please justify.

-          No filler was used for SCC mixtures?

-          Figure 3: why the air test was considered in this study?

-          The reviewer recommends showing the results of CO2 emissions in Figure instead of Table 5.

-          Figure 12: Please extract the maximum stress and area under each mixture's uniaxial tensile test (absorbed energy) and compare the results.

-          General comment: only a few parts of the text discussed the textile within this manuscript. The reviewer recommends explaining more discussion regarding textiles, such as the interaction between textiles and concrete.

Minor editing of English language required.

Reviewer 3 Report

Potential readers will be grateful, if the paper will be supplied by the below mentioned information  and some questions will be answered. 

The paper should be supplied by the clear formulations of goal and tasks of the current investigation. 

Some predictions regarding durability of the structures made of the sea concrete should be done in the current paper.

The paper should be supplied by the information, how admixtures of sea water and exactly, it salinity, influence the mechanical properties of the sea concrete. Are any limitations regarding the using of the sea water and sand for the concrete mixtures? 

Round 2

Reviewer 2 Report

The authors appropriately improved the manuscript.

Minor editing of English language required.